# Outcome of Newborns with Confirmed or Possible SARS-CoV-2 Vertical Infection—A Scoping Review

**DOI:** 10.3390/diagnostics13020245

**Published:** 2023-01-09

**Authors:** Andreea Moza, Florentina Duica, Panagiotis Antoniadis, Elena S. Bernad, Diana Lungeanu, Marius Craina, Brenda C. Bernad, Corina Paul, Cezara Muresan, Razvan Nitu, Raluca Dumache, Daniela Iacob

**Affiliations:** 1Department of Obstetrics and Gynecology, Faculty of Medicine, “Victor Babes” University of Medicine and Pharmacy, 300041 Timisoara, Romania; 2Clinic of Obstetrics and Gynecology, “Pius Brinzeu” County Clinical Emergency Hospital, 300723 Timisoara, Romania; 3Bucharest Emergency Clinical Hospital, 014461 Bucharest, Romania; 4Alessandrescu-Rusescu National Institute for Mother and Child Health, Fetal Medicine Excellence Research Center, 020395 Bucharest, Romania; 5Department of Biochemistry and Molecular Biology, Faculty of Science, University of Southern Denmark, 5230 Odense, Denmark; 6Center for Laparoscopy, Laparoscopic Surgery and In Vitro Fertilization, “Victor Babes” University of Medicine and Pharmacy, 300041 Timisoara, Romania; 7Center for Modeling Biological Systems and Data Analysis, “Victor Babes” University of Medicine and Pharmacy, 300041 Timisoara, Romania; 8Department of Functional Sciences, Faculty of Medicine, “Victor Babes” University of Medicine and Pharmacy, 300041 Timisoara, Romania; 9Department of Neuroscience, Faculty of Medicine, “Victor Babes” University of Medicine and Pharmacy, 300041 Timisoara, Romania; 10Center for Neuropsychology and Behavioral Medicine, “Victor Babes” University of Medicine and Pharmacy, 300041 Timisoara, Romania; 11Department of Pediatrics, Faculty of Medicine, “Victor Babes” University of Medicine and Pharmacy, 300041 Timisoara, Romania; 12Clinic of Neonatology, “Pius Brinzeu” County Clinical Emergency Hospital, 300723 Timisoara, Romania

**Keywords:** pregnancy, SARS-CoV-2, COVID-19, vertical transmission, congenital transmission, placenta, amniotic fluid, umbilical cord

## Abstract

Severe acute respiratory syndrome virus 2 (SARS-CoV-2), the virus that causes 2019 coronavirus disease (COVID-19), has been isolated from various tissues and body fluids, including the placenta, amniotic fluid, and umbilical cord of newborns. In the last few years, much scientific effort has been directed toward studying SARS-CoV-2, focusing on the different features of the virus, such as its structure and mechanisms of action. Moreover, much focus has been on developing accurate diagnostic tools and various drugs or vaccines to treat COVID-19. However, the available evidence is still scarce and consistent criteria should be used for diagnosing vertical transmission. Applying the PRISMA ScR guidelines, we conducted a scoping review with the primary objective of identifying the types, and examining the range, of available evidence of vertical transmission of SARS-CoV-2 from mother to newborn. We also aimed to clarify the key concepts and criteria for diagnosis of SARS-CoV-2 vertical infection in neonates and summarize the existing evidence and advance the awareness of SARS-CoV-2 vertical infection in pregnancy. Most studies we identified were case reports or case series (about 30% of poor quality and inconsistent reporting of the findings). Summarizing the existing classification criteria, we propose an algorithm for consistent diagnosis. Registration: INPLASY2022120093.

## 1. Introduction

In the last two decades, coronaviruses such as severe acute respiratory syndrome coronavirus 2 (SARS-CoV-2), severe acute respiratory syndrome coronavirus (SARS-CoV), and Middle East respiratory syndrome coronavirus (MERS-CoV) have caused the outbreak of various diseases with high contagion and increased risk for human lives [1,2,3].

The ongoing coronavirus disease 2019 (COVID-19) pandemic, caused by SARS-CoV-2, has gained an unprecedented magnitude, generating a global public health crisis that could also have future side effects on future generations. In this context, the scientific community’s effort worldwide has focused on unraveling the different aspects of this new SARS variant, including its structure and mechanisms of action. In contrast, much attention has been given to developing accurate diagnostic tools, drugs, and vaccines to diagnose, prevent, and treat COVID-19 [3,4].

Coronaviruses are members of a family that includes enveloped viruses that replicate in the cytoplasm of host cells, including pets, farm animals, birds, and humans, leading to respiratory and gastrointestinal manifestations [4,5]. Even though SARS-CoV-2 infection is instead associated with the host respiratory system, the lungs are not the only organs that can be affected. By dissemination through the bloodstream, other tissues are susceptible to SARS-CoV-2 infection by the expression patterns of the ACE2 receptor. Consequently, the digestive, neurological, and cardio-vascular systems can be affected, as well as the kidney, liver, and placenta [6,7,8].

SARS-CoV-2 infection’s impact on pregnancy and the effects of the virus on newborns diagnosed with COVID-19 have raised many questions related to the mode of transmission from mother to fetus [6]. However, despite histomorphological and ultrastructural changes in the placentas of COVID-19-positive mothers, only a small percentage of newborns were found to be infected at birth, and no teratogenic effect of COVID-19 infection has been reported [9].

Despite many contradictory discussions and a low number of reported cases since then, the assumption that in utero SARS-CoV-2 vertical transmission is possible has been demonstrated in some published studies [6,10,11,12,13]. How the SARS-CoV-2 virus is transmitted and affects pregnant women and their newborns requires knowledge of the genome and proper use of diagnostic and treatment tools.

We conducted a scoping review with the primary objective of identifying the types, and examining the range, of available evidence of vertical transmission of SARS-CoV-2 from mother to newborn. We also aimed to clarify the key concepts and criteria for diagnosis of SARS-CoV-2 vertical infection in neonates and summarize the existing evidence and advance the awareness of SARS-CoV-2 vertical infection in pregnancy. To attain these latter objectives, we present the medical evidence sources’ synthesis in the context of available knowledge, concepts, and theories in regard to the SARS-CoV-2 infection in neonates.

## 2. General Aspects of the SARS-CoV-2

### 2.1. Genomics of SARS-CoV-2

The 2019 coronavirus disease (COVID-19) etiological agent was initially identified and characterized as a novel beta-coronavirus named 2019-nCoV. However, it was later classified as severeacute respiratory syndrome virus 2 (SARS-CoV-2) by the International Committee on Taxonomy of Viruses based on phylogenetic analysis and genomic structures, being established that the new viruses belong to the family Coronaviridae, sub-order Cornidovirineae, order Nidovirales, and realm Riboviria according to ICTV (International Committee on Taxonomy of Viruses) [14].

This taxonomic order includes viruses with genomes that comprise an extensive replicase gene on the 5′, which covers around two-thirds of the whole genome. On the 3′, genes encoding structural proteins can be found together with accessory genes. In nidoviruses, the replicase gene encodes the replicase-transcriptase polyprotein and contains two ORFs (ORF1a and ORF1b) that get expressed through a ribosomal frameshifting mechanism [15]. The only protein directly translated from the genome is the replicase-transcriptase polyprotein, whereas the rest of the ORFs are expressed through subgenomic mRNAs. The transcription of nested subgenomic mRNAs (sg-mRNAs) from genome complementary negative-sense RNA molecules is a common feature of nidoviruses, and therefore their taxonomic name emanates from the Latin word “nido”, which means “nest” [16].

The size of the SARS-CoV-2 genome is around 30 kb, placing it among the giant genomes within the group of RNA viruses. Likewise, eukaryotic mRNAs have a cap structure at 5′ and a polyadenylated 3′ tail [17]. From 5′ to 3′, the following primary genes can be found in the same order: replicase, S, E, M, and N genes. Accessory genes are located between these genes at specific loci of the 3′, namely the 3a, ORF6, 7a, 7b, ORF8, and ORF10 genes (Figure 1). Interestingly, in coronaviruses, an accessory gene is never placed between the E and M genes [18]. In SARS-CoV-2, additional ORFs have been found, such as ORF3c within ORF3a and ORF9b within the N gene. It is thought to be expressed through a mechanism of “leaky scanning” for translation start sites by host ribosomes [19,20]. The evolutionary origin of the accessory genes is debatable, as they could have been included in the viral genome through intragenomic recombinations or horizontal genetic transfers [21].

Viral proteases enzymatically process the polyproteins encoded by ORF1a and ORF1b after translation to produce non-structural proteins. Within ORF1a, 11 non-structural proteins are included (nsp1–11), whereas five additional proteins are encoded by ORF1b (nsp12–16). The polyprotein 1a (pp1a) has all the non-structural proteins of ORF1a, whereas the polyprotein 1ab (pp1ab) includes nsp1 to nsp10 and the non-structural proteins of ORF1b (Figure 1). Within the coding region of nsp11, there is a “slippery sequence” of 5′-UUUAAAC-3′. When UUA is used as a codon, there is a rise to a downstream termination codon and the generation of pp1a, whereas when UUU is used as a codon, pp1ab is produced [22,23]. It has been indicated that the most critical RNA structure regulating programmed ribosomal frameshifting (PRF) is a pseudoknot downstream of the “slippery sequence”. This pseudoknot can interact with the 40S ribosomal subunit at the entrance channel of mRNA, pause translation, and subsequently induce ribosome slippage [15]. Additional regions that could regulate PRF frequency are the stop codon of ORF1a and an RNA loop, which lies a few bases before the frameshifting site. The stop codon of ORF1a lies five codons after the “slippery sequence”. It has been proposed as an inducer for the pseudoknot formation, whereas the RNA loop has been implicated as an attenuating factor of PRF frequency [22].

The negative sense RNAs, which serve as templates for the transcription of sg-mRNAs, are generated by a mechanism that includes paused synthesis at transcription regulatory sequences (TRS). TRS can be found right after the ORFs of structural and accessory genes [24,25]. At the 5′ end of sg-mRNAs, a typical leader sequence is located, which includes around 70 nucleotides [26,27].

The leader sequence has been indicated as a possible pattern for the initiation of viral mRNA capping. On the other hand, the conserved complementary sequence of the typical leader sequence plays a critical role as a conserved signal for synthesizing positive-sense gRNA and sg-mRNAs [9]. In total, ten sg-mRNAs have been reported, namely sg-mRNA S, 3a, E, M, 6, 7a, 7b, 8, N, and 10 (Figure 1) [28].

### 2.2. Diagnostic Approaches Used to Detect SARS-CoV-2 Infection

Direct and indirect methods have been used to diagnose the SARS-CoV-2 infection.

#### 2.2.1. Viral Isolation Methods

The direct methods involve viral isolation, which is a laborious and relatively slow method of diagnosis. The implementation of faster centrifugation techniques and the use of newly developed cell lines as co-cultured cell lines (multiple cell types) or trans-genic cells (genetically engineered cell lines) have all improved the outcome of this technique [1,3]. In addition, cell cultures can be used together with other assays, such as electron microscopy, to identify the morphology of the virus. Immunofluorescence assays (IFAs) and immunohistochemistry (IHC) use viral antigens for specific identification of a virus [4,5]. But these methods are time-dependent and require special conditions for implementation.

The urgent need to identify and characterize this virus, which demonstrates a high mutational rate and comprehend its immunological and pathophysiological effects on host organisms, prompts the use of new diagnostic techniques and procedures.

Although the gold standard for diagnosing viral infections is represented by the isolation of viral pathogens in cell cultures, numerous direct diagnostic tests that detect the presence of the coronavirus have been applied during this pandemic [3].

#### 2.2.2. Protein-Based Detection Methods

Protein-based detection methods can detect the viral structural proteins and particles of SARS-CoV-2, such as mass spectrometry and biosensors [16].

Detecting the virus at its biological source involves direct diagnostic immunological or molecular methods. Immunological tests detect specific antibodies or antigens. A variety of diagnostic immunoassays exist, including western blots, enzyme-linked immunosorbent assays (ELISA), chemiluminescence immunoassays (CLIA), lateral flow immunoassays (LFIA) and agglutination reactions [4,5,20,25].

Molecular methods, based on the analysis of viral RNA, are multistep procedures that employ a diverse range of specimens, including buccal and NPSs, feces, and bronchoalveolar lavage (BAL) samples. The viral RNA can also be found in blood and other bodily fluids [23,24,25].

#### 2.2.3. Methods Based on the Reverse Transcription Polymerase Chain Reaction

Confirmation of SARS-CoV-2 infection is typically accomplished through nucleic acid amplification tests (NAATs), as the WHO recommends, with the majority of healthcare authorities worldwide employing Real-time reverse transcription polymerase chain reaction (RT-qPCR) in addition to other RT-PCR-based techniques [26].

Multiplex RT-qPCR (capable of targeting different viruses), digital PCR (dPCR), reverse transcription loop-mediated isothermal amplification (RT-LAMP) (which relies on the amplification of specific cDNA sequences to detect SARS-CoV-2) and CRISPR are other approaches used to detect SARS-CoV-2 infection [27,28].

Although methods based on the reverse transcription polymerase chain reaction (RT-PCR) represent the gold standard for diagnosing SARS-CoV-2 infection, these tests do not provide information about the evolution of the infection. For this reason, serology-based tests detect viral antigens or host antibodies [29,30,31,32].

#### 2.2.4. Serology-Based Tests

Anti-SARS-CoV-2 antibodies are detected and quantified to track the progression of the disease and the immune response to vaccination. In addition, serological tests identify active infection by detecting viral antigens in serum, tissues, other body fluids, secretions, or eliminations from infected individuals [33,34,35,36].

However, most research articles report that real-time RT-PCR techniques were previously used to check all biological specimens collected from pregnant women or newborns for the presence of the SARS-CoV-2 virus. Due to their high-throughput specificity and sensitivity, real-time PCR techniques were widely used. Because the number of copies of RNA produced during one amplification cycle increases exponentially and is proportional to the amount of biological material introduced in the reaction—a quantity that depends on viral load in the analyzed sample—this technique can be used to amplify and detect a single copy of some specific genomic sequence. It is also a quantitative technique [37].

In this instance, improperly collected data may yield false-negative results. The use of artificial single-stranded DNA primers and probes, which may bind non-specifically and produce false-positive results, is another limitation of RT-PCR techniques [38].

According to international regulations, diagnostic tests for SARS CoV-2 infection detection must meet specific requirements, including high accuracy, sensitivity, specificity, a low limit of detection, and a quick turnaround time. As a result, other swift, sensitive, and accurate techniques for detecting viral RNA were developed alongside RT-PCR, including LAMP (loop-mediated isothermal amplification) and reverse transcription. Compared to conventional PCR, the RT-LAMP method’s use of 4 to 6 primers increases the selectivity [5,39].

#### 2.2.5. Diagnostic Tools and Their Role in Demonstrating SARS-CoV-2 Vertical Transmission

Presently, evidence-based knowledge in the case of SARS-CoV-2 infection in pregnant women is limited, and it is well-known that clinical trials almost always exclude this category of patients [35,36].

A multidisciplinary, international team of experts proposed some categorization systems to understand vertical transmission and determine the timing of fetal or neonatal infection based on WHO standard recommendations [40].

Despite the many papers published to date, no standardized international definition has yet been established to allow for the comparison of different data from studies that report specifics about COVID-19 infection in pregnancy. Most literature reports refer to a few types of biological samples and diagnostic technologies that have been used to elucidate the timing of vertical transmission of infection from mother to fetus or the neonate.

As recommended by the WHO COVID-19 case definitions, vertical transmission is possible if more than one test for SARS-CoV-2 infection detection in the fetus or neonate is positive after the maternal infection has been confirmed at any point during pregnancy. Because non-sterile samples are more likely to be exposed to cross-contamination, it is advised to use sterile samples for viral detection [26].

The majority of published data regarding infection with the SARS-CoV-2 virus was analyzed from specimens collected from fetuses (such as amniotic fluid and placental fragments) or neonates using NPS testing or other test sites such as rectal or anal swabs, urine, cord blood, or any other fetal or neonatal sources [41,42,43].

Although real-time RT-PCR methods are usually used to detect the presence of viral genetic fragments, other detection techniques can be used to elucidate mother-to-fetus transmissions [42,44,45,46].

Among these, transmission electron microscopy (TEM), immunohistochemistry (IHC), and single-molecule RNA in situ hybridization (ISH) have demonstrated increased specificity for the identification of positive samples [43,46,47,48]. In addition, serological methods have also been used to detect IgM and IgA antibodies against SARS-CoV-2 proteins in neonates. Using these tests, antibodies have been detected less than seven days after birth, indicating most likely fetal infection response in utero, intrapartum, or early postpartum [43,49,50,51,52].

## 3. Maternal-Fetal-Neonatal SARS-CoV-2 Transmission

### 3.1. Vertical vs. Congenital Transmission of SARS-CoV-2

Vertical transmission is the transfer of the pathogen from the mother to the fetus. The infection can occur during pregnancy (via the transplacental route), during birth (when the fetus is in contact with the mother’s reproductive tract), or during breastfeeding. Congenital infections are vertically transmitted from mother to fetus during pregnancy, birth, or breastfeeding. Although the two terms are similar, they are frequently used ambiguously in the literature. Some articles, for example, refer to congenital infections as strict in-utero exposure and vertical infections as strict intrapartum exposure. We used intra-uterine exposure (IUE) and intrapartum exposure (IPE) to avoid ambiguity.

Furthermore, a positive RT-qPCR SARS-CoV-2 result from the neonate’s NPS in the first minutes after birth does not rule out contamination or horizontal transmission, so it should not solely be considered a diagnostic tool for the vertical transmission [53].

It is essential to properly diagnose vertical transmission, as it has both short-term and long-term consequences for the baby. That is why standard classification systems should be followed when analyzing the three types of congenital transmission [54].

### 3.2. Proposed Criteria for Diagnosing Vertical Transmission

#### 3.2.1. Classification Criteria Defined by World Health Organization

According to the World Health Organization (WHO), one must meet three criteria to prove the IUE of SARS-CoV-2 [26]. Firstly, the infection must be confirmed in the mother during pregnancy. Secondly, fetal exposure to SARS-CoV-2 in utero must be evidenced by a positive RT-PCR result in samples such as amniotic fluid or the placenta. If this cannot be performed, specific immunoglobulins A or M in neonatal blood at birth is another sign of intrauterine exposure. Samples from the upper respiratory tract can be collected on the newborn’s first day. Thirdly, the persistence of the infection or an immune response must be documented, either through RT-PCR or the detection of IgA or IgM in the first two days of life. IUE can be classified as confirmed, possible, unlikely, or indeterminate based on these three criteria. The intrauterine transfer is classified only as possible if the persistence of the immune response can be proven from a sample that is not sterile. To differentiate between IUE and IPE, the WHO recommends serial detection of IgM and IgG antibody in the neonate [26]. In cases of intrauterine fetal death, the maternal–fetal transfer of the SARS-CoV-2 virus can be confirmed with the condition that the virus is evidenced in the fetal tissue on RT-PCR techniques or using in situ hybridization methods [26].

#### 3.2.2. Classification System Proposed by the Nordic Federation of Societies of Obstetrics and Gynecology

The Nordic Federation of Societies of Obstetrics and Gynecology (NFSOG)proposed a simpler classification system, described in the paper of Shah et al. [40]. To confirm an intrauterine fetal infection with the SARS-CoV-2 virus, one should have either a positive RT-PCR from an amniotic fluid sample (in the case of a cesarean section) or from umbilical cord blood at the time of birth or a positive RT-PCR from neonatal blood drawn within the first 12 h of delivery. According to him, a positive RT-PCR in an NPS at birth and on the second day of life can only demonstrate IPE in the newborn. To confirm the IUE in a stillbirth, RT-PCR or viral growth from fetal or placental tissue should be performed, or viral particles should be detected using electron microscopy.

### 3.3. Rates and Statistics Concerning the Vertical Transmission

Vertical transmission is a reality, as studies have shown that the SARS-CoV-2 virus is present in newborns’ placentas, amniotic fluid, and umbilical cord [51,55]. The question remains as to how frequently it occurs and what are the potential risk factors for mother-to-fetus virus transmission. There is a paucity of literature when looking for studies that use the proper diagnostic criteria for vertical transmission [56]. In her study of 42 neonates, Sevilla-Montoya found 5 cases (11.9%) where vertical transmission is possible [57]. Kotlyar reports 3.2% (27/936) of neonates with positive RT-PCR NPS within the first 48 h of life and two cases where viral particles were identified at the level of the placenta, hence demonstrating IUE [50]. Another study that involved 70 neonates born from infected mothers concluded that in 5 cases (7.1%), the vertical transmission was considered possible [58]. Upon investigating the presence of IgM antibodies in neonates born from infected mothers, Massalha reports a rate of 3% of vertical transfer [59]. A study that included 14 positive women (7 at delivery) detected one case where the placenta, amniotic fluid, and umbilical cord blood were positive and another with a positive nasopharyngeal aspirate at birth 48 h later [60]. Finally, using NFSOG classification system, Jeganathan reported confirmed vertical transmission in 0.3% of cases, probable vertical transmission in 0.5% of cases, and possible vertical transmission in 1.8% of cases [9].

### 3.4. Intrauterine Fetal Exposure to SARS-CoV-2

#### 3.4.1. Placental Infection with SARS-CoV-2

There is still much to understand about the mechanism of coronavirus transmission from mother to fetus; however, for transplacental transmission to occur, the virus must first be circulating in the bloodstream of the infected pregnant woman. It will enter the placenta’s fetal side through the uterine arterioles before moving on to the chorionic villus and spreading throughout the developing fetus [61]. There may be an association between the duration of viral exposure in utero and neonatal SARS-CoV-2 status. A more extended period of viral exposure may increase the likelihood of neonatal infection [62]. In addition, a high viral load combined with severe inflammation can result in viremia in a neonate [63]. The presence of the angiotensin-converting enzyme-2 (ACE2) receptor, as well as transmembrane serine protease 2 (TMPRSS2), are essential as they promote viral activation in the host cell and, thus, infection [9]. According to some studies, the two receptors are widely distributed in specific cell types of the maternal-fetal interface [10]. Although more abundant in the last trimester of pregnancy, the number of receptors may vary between women, some studies suggesting that there may also be a deficiency in the co-expression of the ACE2 receptor and the TMPRSS2 protease [64,65]. This feature may be attributed to genetic peculiarities and could explain why vertical transmission is so uncommon [66]. Nonetheless, other proteins such as CD147, DPP4, GRP78, L-SIGN, and DC-SIGN may facilitate viral binding and seem to be directly involved in the passage of the SARS-CoV-2 virus through the placenta [34]. Infection of the placenta does not necessarily mean an infection of the fetus, indicating that even though it is not entirely effective, the placental barrier is a significant factor in the low likelihood of COVID-19 vertical transmission [34]. According to Mourad, proteins such as IFITM (Interferon-induced transmembrane protein) 1 and IFITM3 are involved in the trophoblastic immune response and may promote placental protection against SARS-CoV-2 [67]. On the other hand, a homozygous SNP rs12252-C mutation of the IFITM3 increases the risk of SARS-CoV-2 placentitis [68].

Extensive destruction of the syncytiotrophoblast, either through the direct cytotoxic effect or indirectly through circulatory disturbances, can promote the virus’s spread in the villous stroma and its subsequent entry into the bloodstream of the fetus [69].

#### 3.4.2. Fetal Infection with SARS-CoV-2

Regarding fetal coronavirus tropism, studies show that ACE2 and TMPRSS2 are found in the fetus’s smooth, cardiac muscle (with a high density, particularly at the level of cardiomyocytes), lung, and liver. The levels of ACE2 are reported to be especially high in fibroblasts and hepatocytes. ACE2 and TMPRSS2 were also detected in the fetal lungs, both in the epithelial and arterial endothelial cells. This route could explain the possibility of fetal intrauterine lung infection [70]. So far, the brain has been considered an invulnerable organ because of the absence of ACE receptors in the white and grey matter; however, current research indicates that the virus can be found in the choroid plexus of adults as well as fetal brains. From this perspective, the choroid plexus can be an entry point for an invasion of the central nervous system [71].

#### 3.4.3. Intrapartum Fetal Exposure to SARS-CoV-2

Intrapartum exposure can occur after contact with maternal blood, vaginal secretions, or feces. In a study that included 80 women with confirmed COVID-19 infection (RT-PCR from an NPS), 12.5% presented positive vaginal RT-PCR results, and 7.5% had a positive rectal swab [72]. The possibility of neonatal infection during the intrapartum period needs to be considered, mainly if an aspirate was positive after birth. However, the likelihood of vaginally delivered newborns developing COVID-19 has not increased [73,74]. It has been proposed that vaginal microbiota can influence the ascension of the coronavirus, but more research is needed to confirm this [75].

## 4. Review of Available Evidence on Neonatal Outcome in Case of Vertical Transmission

Numerous studies follow the outcome of pregnancies with maternal SARS-CoV-2 infection. Chi et al. reported an 8.8% rate of positive neonates at birth [76]. However, because the criteria for vertical infection are rarely investigated in extensive studies, this type of infection cannot be included or excluded. Clinical data showed a positive outcome in most infants who tested positive for SARS-CoV-2 [77]. While most newborns with a positive SARS-CoV-2 test at birth do not exhibit any clinical abnormalities, some babies do exhibit mild to severe clinical disease. According to Garcia, fever is the most typical symptom, followed by respiratory and gastrointestinal symptoms. Lethargy is the leading neurological symptom that may be present. Although rare (6.8%), cardiovascular symptoms can severely impact the neonate [60].

Ishqeir et al. observed a threefold increase in persistent pulmonary hypertension in patients born to positive mothers [78]. Other studies report shock, arrhythmias, and even thrombosis in positive neonates [79]. Last but not least, a higher stillbirth rate was observed during the pandemic, particularly in mothers with SARS-CoV-2 infection compared to non-infected mothers (5 vs. 3 per 1000 births, *p* = 0.003) [80]. In this review, we will address only cases where the infection is confirmed or possible/probable. Cohort studies have failed to demonstrate vertical transmission [81,82]. At best, they concluded that it was possible [83,84].

### 4.1. Review Approach and Methods

We conducted a scoping review to determine the outcome of newborns where vertical transmission was at least possible according to the WHO and NFSOG criteria. The study was performed according to the PRISMA extension for scoping reviews (PRISMA ScR) guidelines and the Joanna Briggs Institute Reviewers’ Manual for scoping reviews [85,86]. Whenever possible, the search was performed based on the PICO framework: (P) participants—pregnant women; (I) investigated condition—SARS-CoV-2 positive; (C) comparison—SARS-CoV-2 negative; (O) outcome—newborn outcome [87].

The protocol for this scoping review was registered on the International Platform of Registered Systematic Review and Meta-analysis Protocols (INPLASY): unique ID INPLASY2022120093. It is available in full on (https://inplasy.com/inplasy-2022-12-0093/, accessed on 23 December 2022) [88].

Two search engines were used: PubMed/MEDLINE and Google Scholar. The following keywords were used: (‘COVID*’ OR ‘SARS-CoV-2*’) AND (‘vertical transmission’ OR ‘in-utero transmission’ OR ‘congenital transmission’ OR ‘placental infection’). The search period was from 1 January 2020 to 1 November 2022. Reference lists of all identified sources were searched for additional sources. We took into consideration only publications written in English.

No limit or restriction was imposed in regard to the type of study: all types of evidence are to be taken into consideration. Eligible publications comprised the following categories: systematic reviews, case reports and case studies, articles that describe vertical transmission at a molecular level, case–cohort studies, case–control studies, longitudinal cohort studies, cross-sectional studies, descriptive studies, and studies based on surveys. Systematic studies were excluded to avoid the risk of entering a case multiple time (Figure 2).

The studies were identified by two separate researchers who screened the articles and excluded the duplicates in the first stage. Next, abstracts of all potentially relevant papers were individually examined for suitability.

The following inclusion criteria were used to determine the eligible articles:Application of the standard criteria (the WHO or NFSOG criteria) in the attempt to diagnose vertical transmission;Delivery after 20 weeks of gestation;Delivery using strict infection control and prevention practices;Mother–neonate separation at least for 24 h after birth.

All disagreements were resolved by consensus or a third senior researcher was called in to settle.

In determining the type of classification, we followed an adapted table that comprised the WHO and NFSOG criteria (Appendix A). There is a slight discrepancy between the two used classification systems. WHO proposed a classification system that includes confirmed, possible, and unlikely possibilities, while the NFSOG classification system includes confirmed, probable, possible, unlikely, and uninfected cases. To avoid confusion, probable and possible cases classified according to the NFSOG system were grouped under the heading of possible.

Quality assessment was conducted based on the criteria proposed by Murad et al. [89] for the case or case series reports. For the clinical studies, the quality was assessed based on Newcastle–Ottawa Scale Coding Manual [90].

### 4.2. Review Results

Overall, there were 49 case series/reports and three descriptive studies, and a mean score of 3.04 resulted. For the clinical studies, a mean of 5.3 resulted.

Figure 3 shows the assessed quality for the two types of evidence we identified. Full information and scorings for each source of evidence are provided in Appendix A.

The following data were extracted: first author, number of eligible patients in the report, maternal clinical status at the moment of delivery, presence of maternal severe COVID-19 disease, type of vertical transmission according to the WHO or criteria of NFSOG, gestational age at delivery, type of delivery (for live births), neonatal outcome (live birth/stillbirth), presence of any symptoms that could raise the suspicion of SARS-CoV-2 neonatal infection in the first 24 h, neonatal evolution during hospitalization as well as maternal evolution and maternal clinical status at the moment of neonatal discharge.

#### 4.2.1. Possible Association between Vertical Transmission and Adverse Neonatal Outcome

Out of the 75 included cases, there were 32 (42.6%) stillbirths [69,91,92,93,94,95,96,97,98,99,100,101,102,103,104,105,106,107,108,109]. Nineteen neonates (25.3%) presented no symptoms at birth or during hospitalization [10,60,110,111,112,113,114,115,116,117,118,119,120]. Twenty four neonates had symptoms of COVID-19 disease (32%) neonates [52,121,122,123,124,125,126,127,128,129,130,131,132,133,134,135,136]. While most cases (75%) were classified as confirmed vertical transmission, the rest could only be classified as possible vertical transmission (Table 1).

#### 4.2.2. Livebirths Characteristics

In two cases, the livebirth neonates resulted from multiple pregnancies (1 pair of twins and one pair of triplets), and the rest of the neonates resulted from singleton pregnancies. Gestational age at birth ranged from 25 to 40 weeks in the symptomatic newborns and 29 to 40 weeks in the asymptomatic newborns, with a median of 32 in both groups (Table 2).

The Apgar score ranged from 4 to 10 in asymptomatic neonates, while in the symptomatic neonates, the Apgar score ranged from 2 to 10. In both groups, the Apgar score median was 9. Overall, 14 infected newborns were delivered by c-section due to signs of fetal distress. In addition, three neonates were delivered by c-section due to severe maternal COVID disease.

#### 4.2.3. Symptoms of SARS-CoV-2 in Neonates Classified as Confirmed/Possible Vertical Transmission

Nineteen authors described symptomatic neonates in which vertical transmission was either confirmed or possible (Table 3).

Symptomatic neonates experienced most frequently acute respiratory distress in the first 24 h, which evolved in 10 cases in neonatal pneumonia [122,123,126,127,128,129,130,133,136]. Aside from pneumonia, three neonates presented encephalitic symptoms, hypotonia, gastrointestinal symptoms, or mild cutaneous erythema [123,124,127]. Only one case of isolated fever was reported [125]. Hematologic abnormalities were described in five neonates. Zaigham reported a neonate with thrombocytopenia that normalized four days after birth [95]. Kirtsman reported a neonate with neutropenia, and Sukhikh reported a newborn with disseminated intravascular coagulation and congenital anemia [124,128]. Lymphopenia and neutropenia were also reported [135,136]. While four newborns with hematologic abnormalities were classified only as probable IUE, in Sukhikh’sreport transplacental transfer was confirmed (Table 2). One neonate had hypothermia, feeding difficulties, and multiple hypoglycemic episodes [124]. The multi-system inflammatory syndrome was present in one newborn with confirmed IPE [133]. Neurologic abnormalities (axial hypertonia, opisthotonos, and hypotonia) were present in three neonates with confirmed IUE [55,130,137]. COVID-19 was responsible for the death of three neonates on the first, fourth and seventeenth day of life. The first two newborns were delivered in the second trimester of pregnancy at 25, respectively, 26 weeks, while the third was delivered at 34 weeks of gestation. All three cases were diagnosed with congenital pneumonia [122,128,136]. While in the twenty-five-week-old newborn, vertical transmission was confirmed at birth, the 26th-week neonate was initially classified only as probable vertical transmission (Table 2). Postmortem investigation of the neonatal fetal lung demonstrated the SARS-CoV-2 virus in the alveolar macrophages and the pneumocytes (IHC) [122]. The third case of neonatal death remained only a possible vertical transmission as a neonatal autopsy report was not published [136] Except for two newborns who presented neurologic symptoms, clinical evolution improved in all the newborns (respiratory, gastric, and hematologic symptoms have subsided).

## 5. Discussions

The topic of vertical transmission of the SARS-CoV-2 virus is still controversial. Although numerous articles report vertical transmission, cohort studies have failed to demonstrate its evidence [82].

### 5.1. Vertical Transmission, Still an Under-Researched Subject

Even though numerous reviews investigate vertical transmission, only a few, including ours, used standard classification systems [9,138].

Using NFSOG classification, Jeganathan, in his review (2022), identified only 3 cases with confirmed vertical transmission and 17 patients with possible vertical transmission [9]. In this review, we detected ten neonates with confirmed vertical transmission (including Reagan-Steiner’s report) and 14 cases with possible vertical transmission (Table 1). Both classification systems (WHO and NFSOG) were used.

### 5.2. Detecting Vertical Transmission, a Multistep-Time Framed Process

Confirmatory vertical transmission diagnosis can be complicated because it requires collecting sterile samples. Evaluation of the placenta or fetal tissues by a pathologist is recommended but not mandatory. While collecting neonatal blood is relatively simple, collecting the amniotic liquid in sterile conditions can be challenging, especially in the case of vaginal delivery. The procedure entails inserting a sterile needle through the vagina and collecting 10 mL of amniotic liquid prior to the rupture of the membranes. The placental infection could be demonstrated in multiple ways: using RT-PCR, ISH or IHC techniques, or electronic microscopy. When RT-PCR techniques are used, the placenta must also be collected under sterile conditions, a difficult task in case of vaginal delivery. Some authors describe collecting samples deep in the placental tissue as opposed to a simple swab of the fetal and maternal sides of the placenta [95].

### 5.3. Proposed Algorithm for Diagnosing Vertical Transmission

Taking feasibility and cost into account when attempting to diagnose SARS-CoV-2 intrauterine transmission, we propose the following steps: at birth, after cleaning the newborn, a NPS is taken for qRT-PCR. Then, 24–48 hafter birth, a second qRT-PCR should be performed, preferably from neonatal blood. If processing neonatal blood cannot be done due to technical issues, another NPS should be sent for qRT-PCR. This algorithm is partly in accordance with the classification system proposed by Blumberg [49]. If the first NPS swab and neonatal bloodqRT-PCR are positive, the in-utero transmission will be confirmed following WHO criteria [26] (Table 4).

If both NPSs are positive, the intrapartum transmission will be confirmed (according to NFSOG criteria) [40]. A negative NPS at birth with a positive blood sample is classified as confirmed intrapartum transmission (WHO criteria) [26]. Either way, there is a confirmation of vertical transmission. Because the reported incubation period in neonates and children ranges between 2 and 25 days, mother-neonate separation is not required [139]. As recommended by WHO, the two-step diagnosis is imperative as it excludes intrapartum contamination of the fetus with infected maternal blood, vaginal secretions, or feces. Carosso describes such a situation: a newborn with a positive RT-PCR at birth (NPS) and a second negative RT-PCR 36 h later. Before delivery, an NPS was used to confirm maternal infection; however, additional maternal swabs from the vagina, rectus, stool, and colostrum were collected and tested for SARS-CoV-2. Rectal swab RT-PCR was positive [140]. Transient viremia, defined as evidence of early exposure in the absence of neonatal immune response, should also be considered [49].

### 5.4. Vertical Transmission, from the Mechanism of Action to Clinical Aspects

There are still many open questions regarding the mechanism of viral passage in the placenta. According to one theory, a high viral load may indicate a severe infection of the placenta, which would disrupt the normal architecture of the placenta and permit direct transmission to the fetal side of the placenta [63]. Some cite the affinity of the ACE2 receptors for the coronavirus spike protein. However, its actual role in viral transmission is still debatable because some studies indicate that these receptors are naturally upregulated in infected placentas, most likely to reduce inflammation of the placenta [65,141,142]. There is evidence that other proteins (CD147, DPP4, GRP78, L-SIGN, and DC-SIGN) may be more crucial to the transmission process, but the mechanism is unclear [34].

Nine women had a severe form of COVID-19 disease in the case reports we looked at (where vertical transmission was confirmed or at least possible) [10,52,98,101,103,108,114,130,134]. Only in two cases, the newborns showed no signs of infection; the rest were either symptomatic (3) or stillborn (4). Once the virus enters fetal circulation, it has the potential to infect any tissue with ACE2 receptors [30]. However, just as in the general population, the infection does not always result in symptoms. We discovered fifteen case reports of confirmed in-utero infection in neonates who were asymptomatic [10,60,110,111,112,113,114,115,116,117,118,119,120].

The passage of maternal antibodies could explain the absence of symptoms in the newborn. Furthermore, some studies advise against delivery during the acute phase of maternal infection [143]. The majority of the patients in the group we studied were tested upon admission to the hospital during the same period of delivery. Alteration of the fetus’well-being may be directly related to the virus or secondary to a cytokine storm. Overall, four authors reported congenital pneumonia associated with COVID-19 [94,108,122,128]. Confirmation was established postmortem after a histopathologic examination of the lungs. While in two reports, there was an intrauterine death, in the rest, there was a live birth, and neonatal pneumonia was diagnosed in the first days of life. The neonates died on the first, fourth, and seventh days of life, respectively.

### 5.5. Is Screening for Vertical Transmission Necessary?

While the rate of vertical transmission is considered to be very low, we believe that it should be screened in every neonate. Only three years have passed since the appearance of this novel coronavirus, and there is still a lot to discover about the long-term effects of viral infection on the fetus. Yilmaz (2022) observes temporary bilateral hearing loss in neonates born from infected mothers when comparing babies born from non-infected mothers to babies from infected mothers [144]. Ophthalmologic modifications have also been reported. Even though the results are only preliminary, Buonsenso et al. identified abnormal choroidal findings in 15% of neonates [145].

In this review, we concentrated solely on the virus’s direct impact on fetal/newborn health. Cases of chronic or acute fetal distress, although present in our review, were not discussed as they could be the result of placental dysfunction (either through direct invasion by the coronavirus without transplacental transfer or indirectly by the maternal inflammatory markers) or severe maternal COVID-19 disease [137,146,147,148].

## 6. Conclusions

Vertical transmission is still thought to occur on a few occasions, but this could be due to inconsistencies in the application of standard or consistent classification criteria. Although some neonates with confirmed vertical transmission were asymptomatic, poor neonatal outcomes such as neonatal pneumonia, hematologic disorders, feeding difficulties, neurologic abnormalities, and even neonatal death were reported. Infection with the coronavirus can have a significant impact on the human body from an immunological and physiological standpoint.

Long-term sequelae have been reported in the infected adult population, but have to be further assessed and investigated. While some authors speculate that neurological and ophthalmological sequelae may exist in babies born from infected mothers, the long-term effects of in-utero infection remain unknown. Although WHO standard criteria are the most accurate, they are technically challenging to implement and incur high costs. The NFSOG classification system involves fewer steps, but it can be challenging to apply in daily practice, especially if there are no clear hospital guidelines. Serial RT-PCR screening (from NPS) can reveal the possibility of vertical transmission and guide the clinician toward proper surveillance of the future child.

We have synthesized the available evidence on SARS-CoV-2 vertical transmission from the mother to the neonate, and proposed a unified framework for reporting the evidence in a consistent manner. Such consistency is paramount for future systematic reviews and meta-analyses to further the understanding of the underlying mechanisms and subsequent implications. That would be the next step for finding effective clinical approaches and solutions.

## Figures and Tables

**Figure 1 diagnostics-13-00245-f001:**
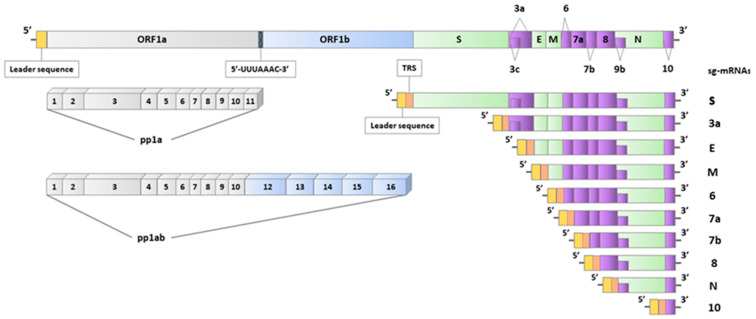
The structure of the SARS-CoV-2 genome and the ten nested sub-genomic mRNAs (sg-mRNAs). The open reading frames (ORFs) are indicated on the genome. The “slippery sequence” (5′-UUUAAAC-3′) is positioned between ORF1a and ORF1b. The polyproteins (pp1a and pp1ab) are expressed by ORF1a and ORF1b and, after post-translational processing, generate 16 non-structural proteins (nsp1-16). A leader sequence can be found at the 5′ of all positive-sense RNA molecules. The transcription regulatory sequences (TRS) are located right after the leader sequence in sg-mRNAs.

**Figure 2 diagnostics-13-00245-f002:**
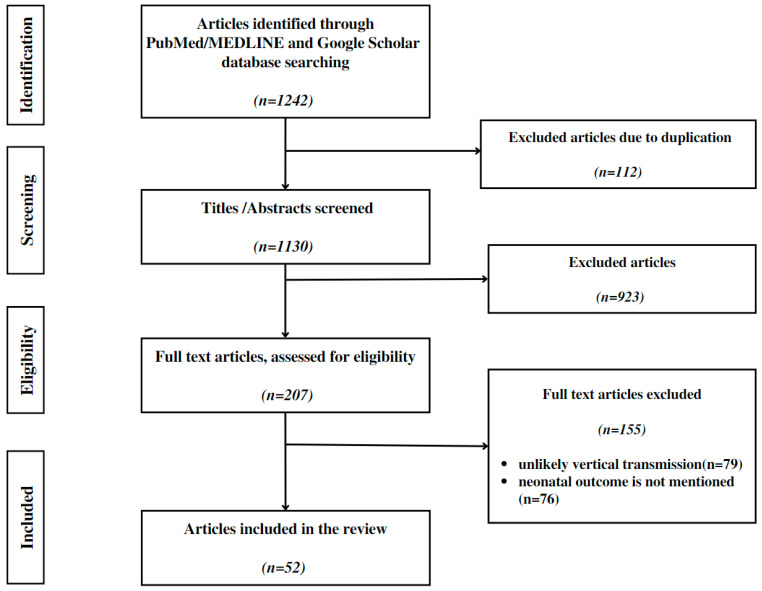
Flow chart of the review process, according to the PRISMA ScR guidelines.

**Figure 3 diagnostics-13-00245-f003:**
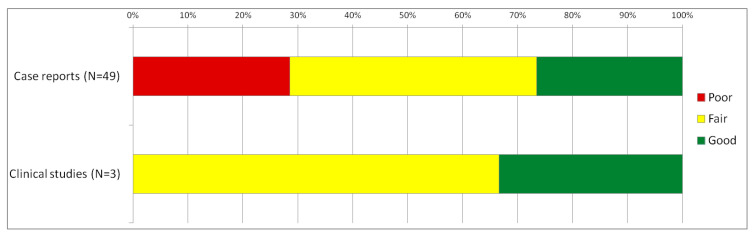
Assessed quality for the identified available sources.

**Table 1 diagnostics-13-00245-t001:** Reported cases with confirmed or possible vertical transmission, stratified by the neonatal outcome.

	Livebirths		Stillbirths
	Asymptomatic	Symptomatic	
Total	19	24	32
C-IUE	15	8	31
P-IUE	4	14	1
C-IPE	-	2	-

Abbreviations: C-IUE, confirmed intrauterine exposure; P-IUE, possible intrauterine exposure; C-IPE, confirmed intrapartum exposure.

**Table 2 diagnostics-13-00245-t002:** Characteristics of livebirths.

	Symptomatic N = 24	Asymptomatic N = 19
Gestational age at the moment of maternal infection, in weeks (median)	32	32
The interval between the moment of maternal infection and birth, in weeks (median)	7	8
Gestational age at delivery, in weeks (median)	34	34
5-min APGAR SCORE (median)	9	9
Birth weight, in grams (median)	2109.5	2109
Caesarian section (c-section)	19 (79%)	18 (94.73%)
Emergency c-section for non-reassuring CTG	8 (33.3%)	6 (18.75%)
Emergency c-section for severe maternal COVID disease	3 (12.5%)	0
Neonatal death	3 (12.5%)	0
Persistence of COVID-19 sequelae after neonatal discharge	2 (8.3%)	0

**Table 3 diagnostics-13-00245-t003:** Essential characteristics of each symptomatic newborn included in the review. The studies are specified in the alphabetical order of the first author’s name.

Author	Type of Vertical Transmission	Gestational Age at Delivery (Weeks)	Neonatal Clinical Outcome	Neonatal Evolution	Case Particularities
**Alzamora** [52]	C-IPE ^b^	33	▪ the newborn was placed on ventilatory support for 12 h, followed by CPAP ▪ mild respiratory difficulty and sporadic cough present ▪ imaging and laboratory testing remain normal	good evolution	▪ severe COVID-19 disease of the mother ▪ c-section due to maternal respiratory compromised status
**Boncompagni** [129]	C-IUE ^a^	35	▪ pneumonia	good evolution	▪ c-section for non-reassuring cardiotocography
**Choobdar** [134]	P-IUE ^a^	31	▪ mild hypotonia ▪ several episodes of apnea	good evolution	▪ maternal death due to severe COVID disease ▪ c-section for mother with critical COVID-19 disease
**Disse** [132]	P-IUE ^a^	28	▪ respiratory distress in DOL 1 ▪ X-ray: pulmonary interstitial emphysema	good evolution	▪ pregnancy with trichorionic triplets ▪ c-section for premature rupture of membranes in a trichorionic triplet pregnancy
	P-IUE^a^	28	▪ respiratory distress in DOL 1 ▪ X-ray: pulmonary interstitial emphysema	good evolution	▪ only fetuses A and C are presented, as investigations for fetus Bare are not complete to diagnose IUE in fetus B
**Ergon** [136]	P-IUE ^a^	34	▪ borderline lymphopenia and thrombocytopenia ▪ X-ray: bilateral ground-glass opacities ▪ intubation required ▪ acute respiratory distress on DOL 6 ▪ persistent pulmonary hypertension	neonatal death in DOL 17	▪ c-section maternal vaginal bleeding
**Facchetti** [127]	P-IUE ^a^	37	▪ developed fever, breathing difficulty, vomiting, abdominal distension, hypotonia, and mild erythemacutaneous ▪ X-ray: interstitial pneumonia - mechanical ventilation was used	good evolution	▪ induced vaginal delivery due to maternal COVID-19 pneumonia ▪ particles morphologically consistent with coronavirus were localized in fetal circulating mononuclear cells
**Farhadi** [130]	C-IUE ^b^	32	▪ respiratory distress ▪ X-ray showed a pneumomediastinum/pneumothorax ▪ chest drainage was performed ▪ lung CT scan revealed patchy shadows in the peripheral parts of the lung	good evolution	▪ c-section for mother with critical COVID-19 disease
**Rebello** [131]	P-IUE ^a^	34	▪ X-ray: diffuse granular opacities but without atypical pulmonary condensation	good evolution	▪ vaginal delivery
**Favre** [137]	C-IUE ^b^	29	▪ abnormal neurological examination	abnormal neurological examination DOL 54	▪ c-section for non-reassuring cardiotocography ▪ viral lineage B.1.221 ▪ severe neurological injury on computer tomography
**Kirtsman** [124]	P-IUE ^a^	35	▪ the neonate presented neutropenia and hypothermia, poor feeding and hypoglycemic episodes	good evolution	▪ C-section due to abnormal maternal coagulogram (patient known with familial coagulopathy),
**Lorenz** [123]	C-IPE ^b^	40	▪ refractory fever (38.6 °C) on DOL 1 ▪ encephalitic symptoms on DOL 2 ▪ X-ray: bilateral pneumonia DOL 10	good evolution	
**Marzollo** [126]	P-IUE ^a^	38	▪ clinical signs of COVID-19 infection present ▪ X-ray: increase of interstitial markings on the base of the left lung	good evolution	▪ induction of labor for severe idiopathic thrombocytopenia in the mother and non-reassuring cardiotocography
**D. C. Ng** [135]	P-IUE ^a^	29	▪ respiratory distress on DOL 1 ▪ neutropenia, lymphopenia and thrombocytopenia on DOL 3	good evolution	▪ maternal pneumonia ▪ c-section for impending rupture of maternal colon cancer
**Parsa** [125]	C-IUE ^b^	37	▪ fever that resolved with conventional therapy	good evolution	▪ c-section due to scarred uterus ▪ maternal COVID-19 pneumonia,
**Reagan-Steiner** [122]	P-IUE ^a^	25	▪ severe pneumonia	neonatal death in DOL 4	▪ c-section for preeclamp-sia with severe features ▪ histopathologic evi-dence of bronchopneu-monia ▪ SARS-CoV-2 RNA pre-sent in the myocardium and liver
**Shaiba** [133]	C-IPE ^a^	32	▪ respiratory distress syndrome ▪ X-ray: bilateral ground glass appearance with bilateral haziness on DOL 1 ▪ multi-system inflammatory syndrome on DOL 4	good evolution	▪ c-section for abortion placentae
**Sukhikh** [128]	C-IUE ^b^	26	▪ congenital pneumonia ▪ congenital anemia ▪ disseminated intravascular coagulation in DOL 1 ▪ ultrasound: antenatal intraventricular hemorrhage grade 3 on the right side at the stage of cyst formation and cardiomegaly	neonatal death in DOL 1	
**Vivanti** [55]	C-IUE ^b^	35	▪ neonate presented with neurological manifestations: difficulty in feeding, axial hypertonia and opisthotonos	good evolution	▪ c-section for non-reassuring cardiotocography ▪ maternal viremia present ▪ at two months of life, showed a further improved neurological examination
**Vivanti** [115]	P-IUE ^a^	40	▪ transient tachypnea ▪ required CPAP for three days;	good evolution	
**Zaigham** [95]	P-IUE ^b^	34	▪ thrombocytopenia normalized on DOL 4	good evolution	▪ c-section for non-reassuring cardiotocography
	P-IUE ^a^	34	▪ no spontaneous breathing directly after birth. CPAP for 24 min.	good evolution	▪ c-section for non-reassuring cardiotocography
	P-IUE ^a^	29	▪ no spontaneous breathing after birth ▪ respirator required, followed by CPAP	good evolution	▪ c-section for non-reassuring cardiotocography

^a^ according to WHO criteria [26]; ^b^ according to NFSOG classification [40]; Abbreviations: A-asymptomatic; C-IPE, confirmed intrapartum exposure; C-IUE, confirmed intrauterine exposure; CPAP, continuous positive airway pressure; DOL, day of life; NS, not specified; P-IPE, possible intrapartum exposure; P-IUE, possible intrauterine exposure; S, symptomatic.

**Table 4 diagnostics-13-00245-t004:** The proposed algorithm in diagnosing vertical transmission.

qRT-PCR at Birth Nasopharyngeal	qRT-PCR at 24–48 h		
	Neonatal Blood	NPS	
Positive	positive		C-IUE ^a^
Positive		positive	C-IPE ^b^
Negative	positive		C-IPE ^a^
Negative		positive	C-IPE ^a,^*

* with thecondition of a third positive NPS between the 3rd and 7th day of age. ^a^ according to WHO criteria [26]; ^b^ according to NFSOG Classification [40]; Abbreviations: C-IUE, confirmed intrauterine exposure; C-IPE, confirmed intrapartum exposure.

## Data Availability

Not applicable.

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
