# Peer review of "Outcome of Newborns with Confirmed or Possible SARS-CoV-2 Vertical Infection—A Scoping Review"

_diagnostics, 2023, doi:10.3390/diagnostics13020245_

Round 1
Reviewer 1 Report
the topic is interesting and the arguments are well presented.
The main weakness of this paper is the lack of a methods section.
Actually, it is a review but does not specify how it has been conducted. if it is systematic or not, nor if the protocol was registered in advance.
Electronic scientific databases consulted for the review are not specified. The same for the search strategy. Neither was the time during which the review has been conducted. Please add.
Moreover, included articles are not assessed in terms of risk of bias. Please add.
Please, also specify in the title the type of work performed.
Reviewer 2 Report
Dear author
My main concern is
Is this a review about neonatal SARS-2 transmission os a descriptive collection of clinical cases or soemthing else?
If the manuscript is a calssic review my advice is to rewrite it according to that. To expose and comment the new findings in the field.
If it is a collection of clinical cases, then to rewrite according to Cochrane rules.
If it is soemthing, please to be specific. Sometime whe the literature is lacking of references and study it is possible to admit as a review a mix between a exposure of clinical cases toghether a review of the topic.
Once you define this terms, It will be a pleasure to reviewa properly the manuscript.
In advance, to pay atention to concpets about diagnosis such as mass spectrometry, TEM o even some comments about RT-PCR..
The similarities with the order of Nidovirales is absolutly vast, is an order ...not a family, does not? MAybe the author shou shape this litle details for the new review.
Thanks
Reviewer 3 Report
In this review article, the authors intend to clarify the impact of SARS-CoV-2 infection during pregnancy, particularly its effect on newborns.
Several suggestions:
1. Line 49, please add [Coronavirus] after [Severe Acute Respiratory Syndrome (SARS)] to (SARS-CoV}.
2. Please add references after the sentence in lines 58, 69, 72, 491, 492.
3. Line 95, [5’ to 3’] rather than [5 to 3] is suggested.
4. Line 143, why virus isolation is an indirect method?
5. Line 147, [expression] is not the right term for the CPE?
6. Line 175, please delete [Real-time]. [real-time] is the same meaning as [quantitative].
7. Lines 183-184 [and also 186-187], why serology based tests were used to detect viral antigens?
8. Line 201, please add the reference after [As suggested by the WHO COVID-19 case definitions].
9. Line 210, [Real-time RT-PCR] rather than [Real-time PCR] is suggested.
10. Lines 253-260 [also in lines 320-328], [RT-PCR] rather than [PCR] is suggested.
11. Line 290, [transmembrane serine protease 2] is the official name for TMPRSS2.
12. Please check the sentence [Controlled studies have failed to demonstrate vertical transmission] in line 346.
13. Table 2, it is suggested that [weeks] is added after [Gestational age at delivery].
14. Line 368, please add [c-section] after [cesarian section].
15. Line 427, please add [NPS] after [nasopharyngeal swab]. Then, use NPS rather than nasopharyngeal swab in the sections of Discussion and conclusion.
16. Line 471, please put a number before [st].
Round 2
Reviewer 1 Report
I am satisfied with the changes provided
Reviewer 3 Report
The authors have addressed the issues I have raised previously in this revised manuscript.